# Attribute-Based Encryption in Securing Big Data from Post-Quantum Perspective: A Survey

Zulianie Binti Jemihin [1], Soo Fun Tan [1,*] and Gwo-Chin Chung [2]

1 Faculty of Computing and Informatics, University Malaysia Sabah (UMS), Kota Kinabalu 88400, Malaysia
2 Faculty of Engineering, Multimedia University, Cyberjaya 63100, Malaysia
* Correspondence: soofun@ums.edu.my

**Abstract:** Attribute-based encryption (ABE) cryptography is widely known for its potential to solve the scalability issue of recent public key infrastructure (PKI). It provides a fine-grained access control system with high flexibility and efficiency by labeling the secret key and ciphertext with distinctive attributes. Due to its fine-grained features, the ABE scheme is a protection layer in securing users' data and privacy in big data processing and analytics. However, quantum computing, new technology on the horizon that will transform the security and privacy environment, has begun to appear. Like the conventional ABE schemes, present cryptography is not excluded from the impacts of quantum technology as they are not made to be quantum-resistant. While most recent surveys generally touched on the generic features of attribute-based encryption schemes such as user revocation, scalability, flexibility, data confidentiality, and scope in pairing-based ABE schemes, this survey investigated quantum-resistant ABE schemes in securing big data. This survey reviews the challenges faced by the recent ABE cryptography in the post-quantum era and highlights its differences from the conventional pairing-based ABE schemes. Subsequently, we defined the criteria of an ideal quantum-resistant ABE scheme. Additionally, existing works on quantum-resistant ABE schemes are reviewed based on their algorithms design, security and functionalities. Lastly, we summarized quantum-resistant ABE schemes' ongoing challenges and future works.

**Keywords:** attribute-based encryption; post-quantum cryptography; public key infrastructure





## 1. Introduction

Big data analytics has attracted much attention recently in various industries [1,2]. Its data collection and analytics automation abilities provide a holistic picture of business decision-making and enhanced insight into the manufacturing process, risk management, and customer relationship management. Today, a new alternative is available for organizations to outsource big data to third-party cloud service providers. Security threats and privacy concerns are critical for big data analytics and management [2,3]. Traditionally, symmetric encryption and Public Key Infrastructure (PKI) protect data confidentiality during transmission and storage. However, distributing and managing different pairs of keys became more complex and challenging when the number of users increased, particularly in big data analytics and outsourced cloud computing environments. The lack of scalability and single-point attack issues urged practical security solutions to solve the performance bottleneck of recent PKI [1–4].

Attribute-based Encryption (ABE) promises a practical method to solve the current PKI's bottleneck performance in controlling fine-grained user access [2,4]. ABE was derived from Attribute-Based Access Control (ABAC) concepts and hybridized with conventional public-key encryption. Users can recover encrypted data correctly if the users pose a set of authorized attributes (e.g., genders, roles, ages, etc.) that match the predefined access control policies either in the ciphertexts or secret key. ABE's fine-grained access properties authorize multiple users to access the shared secret data making it a flexible access control

mechanism, confidential information sharing and exchange in recent advanced big data analytics and outsourced cloud computing. ABE scheme has attracted much research attention over the past decades. The survey on the current state of the art of ABE is summarized in Table 1.

**Table 1.** Chronological Summary Attribute-Based Encryption (ABE) Surveys.

| Year | Reference | Quantum Perspective | Application Scenario | Remarks |
|---|---|---|---|---|
| 2014 | Balamurugan and Venkata [5] | × | Cloud application | A general review and discussion on ABE in securing cloud application |
| 2014 | Qiao et al. [6] | √ | General application | General Comparison of ABE features and quantum-based ABE literature limited to 3 articles |
| 2016 | Liu et al. [7] | × | Cloud data storage | A detailed technical review for each selected ABE scheme, however, limited to 4 articles |
| 2017 | Mhatre et al. [8] | × | Health records in cloud storage | A very general non-technical review and literature limited to 13 articles |
| 2018 | Rachman [9] | × | File Storage in the cloud | A very brief non-technical survey and literature limited to 8 articles |
| 2019 | Edemacu et al. [10] | × | Collaborative eHealth | A detailed technical review on bilinear pairing-based ABE |
| 2019 | Al-Dahhan et al. [11] | × | General cloud application | A general review of single and multi-authority CP-ABE schemes |
| 2019 | Sun [12] | × | General cloud application | A detailed technical review on ABE and searchable encryption |
| 2020 | Zhang et al. [2] | × | Cloud Storage application | A technical review on (ABE) for cloud computing access control |
| 2021 | Oberko et al. [4] | × | General application | Focus on Multi-Authority Attribute-based encryption (MA-ABE) schemes. |

Previous surveys scoped their application scenario in controlling access to cloud computing platforms or cloud [2,5,7–9,11,12], whereas Edemacu et al. [10] focused ABE on securing collaborative eHealth. Most of the surveys [5,6,8,9,11,13,14] generally touched on the general features of attribute-based encryption schemes such as user revocation, scalability, flexibility, data confidentiality, etc. and lacked a technical review and comparison of the ABE algorithm design. Moreover, these surveys primarily focused on conventional ABE schemes that were constructed based on pairing-based cryptography, such as Bilinear Diffie–Hellman (BDH), Decisional Bilinear Diffie–Hellman (DBDH), and q-type assumptions are insecure against quantum attacks [15,16]. This paper investigates the recent ABE scheme in securing big data in the cloud environment from the post-quantum perspective. The contributions of this paper are:

- First, we investigate the quantum-resistant attribute-based encryption characteristic and how it differs from conventional attribute-based encryption;
- Then, we discuss evaluating an ideal quantum-resistant ABE scheme that can be adapted to secure big data processing;
- We provide a comprehensive review of the recent quantum-resistant ABE schemes, divided into Key-Policy Attribute-Based Encryption (KP-ABE) schemes and Ciphertext-Policy Attribute-Based Encryption (CP-ABE). We reviewed and compared their al-

gorithm design, access structure, hardness of security assumptions, threat model, revocable features, resistance to collusion attack and ciphertext indistinguishability;

- Lastly, we highlight the quantum-resistant ABE scheme's ongoing challenges and future trends.

The rest of this article is organized as follows. Section 2 discusses the characteristics of quantum-resistant ABE schemes and highlights how it differs from conventional ABE. Section 3 defines the criteria of an ideal quantum-resistant ABE scheme. Subsequently, a detailed technical review of quantum-resistant ABE schemes is presented in Section 4. Section 5 discusses the ongoing challenges and highlights the ABE scheme's future direction. Lastly, Section 6 concludes the paper.

## 2. Quantum-Resistant Attribute-Based Encryption Characteristics and How It Is Different from Recent Attribute-Based Encryption

The development of PKI can be traced back to the early 1970s and evolved continuously to secure communication in the world wide web era. PKI is fundamentally constructed based on modern algebraic number theory approaches, and their hardness of security assumptions has relied on computational security approaches. For instance, the hardness of the RSA algorithm is constructed based on an integer factorization problem, in which the cost and time of breaking it exceed the value and useful lifetime of encrypted data [17,18]. However, the recent advancement of quantum computing exploits the quantum mechanics theory to tackle mathematical problems traditionally intractable to solve by modern computers. The emergence of quantum computing has raised security concerns on current PKI. Shor's algorithm [18] that could factor a large prime number with a polynomial time of complexity $O$ (log $N$) resulted in the RSA algorithm no longer being secure in the quantum era. Subsequently, Grover's quantum search algorithm [19] can achieve quadratic-time ($O(\text{sqrt}(N))$) compared to a classical algorithm with $O(N)$ time complexity lessening the hardness of modern elliptic curve and pairing-based cryptography [20]. This section highlights the characteristics of the quantum-resistant ABE scheme and how it differs from the recent ABE scheme.

The idea of Attribute-Based Encryption (ABE) evolved from Identity-Based Encryption (IBE), in which a trusted authority generates users' private keys based on a collection of descriptive attributes containing the user's private keys and encrypted information. A specific key can only be decoded if the key's properties match the encrypted data. Table 2 summarizes the difference between modern ABE and quantum-resistant ABE.

**Table 2.** The difference between quantum-resistant ABE and recent ABE.

| Concerns | Recent Attribute-Based Encryption | Quantum-Resistant Attribute-Based Encryption |
|---|---|---|
| Algorithm constructions | Pairing-based cryptography, Elliptic curve cryptography, RSA algorithm | Lattice-based problem approach |
| Computational hardness assumption | Bilinear Diffie–Hellman (BDH), Decisional Bilinear Diffie–Hellman (DBDH), and q-type | Shortest vector problem (SVP), Learning with Error (LWE), Ring-Learning with Errors (R-LWE), Nth Degree Truncated Polynomial Ring Units (NTRU) |
| Storage Efficiency | Bit storage *2n n*-bit strings | Bit storage one *n*-bit string |
| Processor Efficiency | Depends on the throughput of the algorithm. | Fast with optimization techniques |
| Transmission Bandwidth Efficiency | Bandwidth depends on computational power | Ranging from 0.6–2.2 KB |

Recent modern ABE algorithms are mainly constructed from elliptic curve cryptography and pairing-based cryptography, including the RSA algorithm's integer factorization

problem, Diffie–Hellman algorithm's discrete logarithm problem, and the elliptic curve discrete logarithm's. These number theory-based approaches are hard in classical computation models; however, neither is hard in quantum computation models, e.g., Shor's algorithm [17] and Grover's algorithm [19]. Even though elliptic curve cryptography is mostly used to construct ABE due to its features, e.g., smaller ciphertexts, keys, and signatures, and faster generation of keys and signatures, it is still weak against quantum attacks [13]. Hence, it is important to derive quantum-resistance constructions with the security base of a cryptographic design on a computational problem that is not known to be solved easily by quantum algorithms.

A few areas were identified that can fulfill this requirement: hash-based cryptography, multivariate cryptography, isogeny-based cryptography, code-based cryptography, and lattice-based cryptography [14–16]. Lattice-based cryptography stands out the most among this quantum-resistant cryptography to address the RSA and elliptic curve discrete logarithm's hardness issues [13]. It promises the alternative algorithm design of ABE schemes with its secure cryptographic complexity assumptions from average-case Learning with Error (LWE) to worse-case Shortest Vector Problem (SVP) [21,22]. SVP is the most basic hardness assumption in lattice-based cryptography, but this assumption is easy to solve if one has a good basis [22]. Whereas LWE is an average-case problem that can handle quantum attacks better than SVP, LWE suffers from quadratic key sizes and computational times in the lattice dimension. Hence, it affects the efficiency of ABE scheme implementation [16]. On the other hand, Nth Degree Truncated Polynomial Ring Units (NTRU) depend on the difficulty of factorizing polynomials into the quotient of two polynomials, which makes it the most practical lattice-based cryptography and secure algorithm that can resist quantum attacks [13,21,23].

Moreover, the quantum-resistant ABE is built with bit storage of one $n$-bit string. Lattice-based cryptography can run faster than conventional cryptography, such as RSA, and can be implemented on low-power devices with 8-bit microcontrollers. For example, implementing recent R-LWE-based encryption [22] on an 8-bit AVR microcontroller can complete encryption within two million cycles. In comparison, the RSA-1024 performance needs more than twenty-three million cycles for the same encryption process. Other candidates of post-quantum cryptography, such as code-based cryptography, can be implemented, resulting in even better performance related to computational efficiency. Still, it requires a larger size for keys and ciphertexts. In practical implementation, bandwidth could be a bottleneck over communication links where packet loss rates are more than 3–5%. When comparing the conventional ABE and quantum-resistant ABE, the quantum-resistant ABE generally provides better bandwidth due to the size of keys and ciphertexts [16].

Therefore, looking at the characteristics of quantum-resistant ABE schemes, it is undeniable that the quantum-resistant ABE scheme is feasible in protecting against quantum attacks. Investigating a quantum-resistant ABE scheme is imperative to ensure the future viability of cryptographic protocols in large-scale quantum computers.

## 3. Evaluation of Ideal Quantum-Resistant Attribute-Based Encryption

Researchers have always tried to invent an ideal ABE scheme to keep up with the pace of current advanced technologies of big data processing and cloud environments. Several design requirements for constructing an ideal quantum-resistant ABE scheme are summarized as follows:

- **Algorithm design**: The design of quantum-resistant ABE schemes should be able to resist quantum attacks. At the same time, performance should not compensate for achieving a more robust quantum-resistant ABE scheme. It is also important to choose an algorithm that can be easily implemented into various devices [22].
- **Access structure**: It is also known as access policy, which is usually expressed as a circuit over a set of attributes. Access structure controls who can decrypt ciphertext [24] and generally can be categorized into monotonic access structure, non-monotonic access structure, and hidden access structure. The monotonic access structure is widely

used in ABE. It comprises AND, OR and threshold gates and leaves that describe attributes. In contrast, a non-monotonic access structure uses NOT gates, including negative key access and generation constraints. The hidden access structure allows the data owner to hide the access structure and encrypt it for secure communication [25,26].

- **Hardness assumption**: Quantum-resistant ABE scheme is hard in the quantum computational model, primarily derived from fundamental lattice-based problems, including the shortest vector problem (SVP) and closest vector problem (CVP). The hardness of learning with error (LWE) enjoys a worst-case lattice problem as the SVP and shortest independent vector problem (SIVP); however, it suffers from the quadratic overhead problem in computation times and key size. Implementing quantum-resistant ABE requires a practical hardness problem in the lattice. R-LWE in ideal lattice enjoys smaller storage and faster operations, thus promising another alternative towards a practical ABE scheme in supporting real-world industry [13,21,23].

- **Threat model**: Like modern ABE schemes, the threat model of quantum-resistant ABE schemes is analyzed using selective and adaptive models. In the selective model, also called a non-adaptive model, the attacker must choose which challenge attribute to attack before accessing the ABE scheme's public parameters or any of the keys. Whereas in the adaptive security model (also known as full security), the challenge attribute can be chosen at any time, even after the attacker obtains the public parameters and decryption keys. When the number of parties is super-logarithmic, the adaptive security model is strictly more robust than the selective security model. In a practical situation, attackers usually break into a system during computation based on the partial information they gathered beforehand. Thus, adaptive security seems to better present realistic security threats and provide a security guarantee [27].

- **Ciphertext indistinguishability**: Similar to the conventional ABE scheme, the quantum-resistant ABE scheme should validate under IND-CPA and IND-CCA. IND represents the goal of security which is indistinguishable. Likewise, CPA and CCA represent the strength of the attack, whether it is a passive adversary or an adaptive chosen ciphertext attack [26]. Most quantum-resistant ABE schemes were proved to be IND-CPA secure and IND-CCA secure. A cryptosystem being IND-CCA1 secure implies that it is also IND-CPA secure. Subsequently, the IND-CCA2 secured also implies IND-CCA1 secured [28,29].

- **Collusion resistant**: Users should not combine their private keys with each other to obtain unauthorized data. Thus, it must be preserved as polynomials or random integers that are unable to be deciphered simply by mixing user attributes [30]. In a multi-authority ABE scheme, the total number of users must not exceed the number of attribute authorities to prevent collusion attacks [10].

- **Revocable**: An ideal quantum-resistant ABE scheme should address the user revocation and attribute revocation. User revocation is a mechanism to auto revoke permissions if any user leaves the system. The revoked user lost authorization, and he or she cannot decrypt the data because access rights were forbidden [31]. The user revocation in quantum-resistant ABE schemes can be categorized as direct revocation and indirect revocation. Direct revocation occurs when senders specify the revocation list while encrypting the message and has the advantage of not requiring a key update phase for all non-revoked users engaging with the key authority. In contrast, indirect revocation is enforced by a trusted key authority that regularly publishes key update materials in such a way that only non-revoked users can update their keys, thus rendering revoked users' keys worthless [10,30] and does not require senders to be aware of the revocation lists.

## 4. Quantum-Resistant ABE Scheme and Recent Works

The concept of the ABE scheme was derived from Sahai and Waters' Identity-Based Encryption (IBE), which was initially described in EUROCRYPT 2005. By using a collection of attributes set as a public key and associating it with either the user's secret key (known

as Key Policy-ABE, KP-ABE) or ciphertext (also known as Ciphertext Policy-ABE, CP-ABE), the ABE allows fine-grained control access to encrypted data. Recent works of quantum-resistant KP-ABE and CP-ABE schemes are further reviewed and discussed as follows.

### 4.1. KP-ABE Schemes

A message is encrypted with a set of attributes in the KP-ABE scheme, and the user's private keys are linked to an access structure. If the access structure in the user's private key can satisfy the encrypted data's attributes, the message can be recovered successfully [2,4,10–12,16,17,30–40]. Generally, the KP-ABE scheme is constructed based on four basic algorithms: setup, key generation, encryption, and decryption, as summarized below.

- Setup ($k \rightarrow pp, mk$): Takes a security element, $k$ and outputs the master key, $mk$ and the public keys, $pp$.
- Key Generation ($\tau$, $mk \rightarrow sk$): Takes the access policy, $\tau$ and the master key, $mk$ and outputs the user's private key, $sk$, corresponding to attributes in the access policy.
- Encryption ($M$, $S \rightarrow CT$): Takes message, $M$ as input the data, a set of attributes, $S$ and outputs a ciphertext, $CT$ associates with the attribute set $S$.
- Decryption ($CT$, $sk$, $pp \rightarrow M/\perp$): Takes the ciphertext, $CT$ as input, the user's private key, $sk$ and the public keys, $pp$ to recover the encrypted message, $M$. The decryption is successful if and only if ciphertext attributes satisfy the access structure in the user's private key. Otherwise, the algorithm outputs $\perp$.

As much as the KP-ABE scheme provides fine-grained access control, it does have a few significant disadvantages, including (i) high communication costs due to each attribute having its negative version in the system; (ii) senders have less control over who can decrypt as the access structure is built into the users secret key; and (iii) insecure data confidentiality by a trusted authority that handles secret keys and subjected to collusion attacks [10,38]. Table 3 summarizes the recent works of quantum-resistant KP-ABE schemes.

**Table 3.** Survey of Quantum-Resistant KP-ABE Schemes.

| Year | Ref. | Algorithm Design | Access Structure | Hardness Assumption | Threat Model | Ciphertext Indistin-guishability | Collusion Resistance | Revocable |
|------|------|------------------|------------------|---------------------|--------------|----------------------------------|----------------------|-----------|
| 2013 | Boyen [41] | Lattice-based | LSSS | LWE | Selective | IND-CPA | No | - |
| 2015 | Boyen and Li [42] | Lattice-based | Boolean | LWE | Selective | IND-CPA | No | - |
| 2017 | Kuchta and Markowitch [43] | Lattice-based | LSSS (Threshold gate) | LWE | Selective | IND-CCA | Yes | - |
| 2017 | Tan and Samsudin [44] | Lattice-based | LSSS (Threshold gate) | R-LWE | Selective | IND-CPA | No | - |
| 2018 | Zelin [45] | Lattice-based | Tree | LWE | Selective | IND-CPA | Yes | - |
| 2018 | Dai et al. [24] | Lattice-based | Boolean circuit with AND and NAND gates | R-LWE | Selective | IND-CPA | No | - |

**Table 3.** *Cont.*

| Year | Ref. | Algorithm Design | Access Structure | Hardness Assumption | Threat Model | Ciphertext Indistin-guishability | Collusion Resistance | Revocable |
|------|------|------------------|------------------|---------------------|--------------|----------------------------------|----------------------|-----------|
| 2018 | Zhao and Gao [46] | Lattice-based | LSSS (AND and OR gates) | LWE | Selective | IND-CPA | No | - |
| 2018 | Yu et al. [47] | Lattice-based | Tree | Decision R-LWE | Selective | IND-CPA | No | - |
| 2019 | Liu et al. [48] | Lattice-based | LSSS (Threshold) | LWE | Selective | IND-CPA | No | - |
| 2020 | Liu et al. [49] | Lattice-based | LSSS (AND, OR and Threshold gates) | LWE | Selective | IND-CPA | No | - |
| 2021 | Luo et al. [50] | Lattice-based | Boolean | LWE | Selective | IND-CPA | No | User-level |
| 2021 | Pal and Dutta [51] | Lattice-based | Boolean | LWE | Adaptive | IND-CCA | No | - |

The first idea of constructing the KP-ABE scheme from lattice-based cryptography was initiated by Boyen [41] in 2013. The KP-ABE scheme was designed based on the hardness of the LWE problem. Subsequently, Boyen and Li [42] further enhanced the scheme to support finite automata with bounded input length from lattices. Kuchta and Markowitch [43] scoped their threshold gate KP-ABE scheme in supporting multiple cloud servers. Zelin [45] applied a tree access structure in constructing the access policy. Tan and Samsudin [44] solved the computation overhead of the previous LWE-based KP-ABE scheme by extending it to the hardness of the Decisional R-LWE problem. They also hybridize the KP-ABE with homomorphic encryption to support the multi-user cloud environments. Dai et al. [24] investigated the practical implementation of R-LWE based KP-ABE scheme using the PALISADE library. Zhao and Gao [46] enhanced LSSS in supporting the access structure of the KP-ABE scheme; however, the number of secret keys increased exponentially. Yu et al. [47] extended tree structure to support AND, OR and threshold gates as LSSS matrix in the KP-ABE scheme. Liu et al. [48] extended KP-ABE in a keyword-searchable context, whereas Liu et al. [49] focused on addressing the leakage of shared master keys in LSSS. Luo et al. [50] applied proxy re-encryption concepts in solving the forward and backward secrecy of the KP-ABE scheme. Pal and Dutta [51] further extended the KP-ABE scheme in supporting functional encryption. Generally, the algorithm design of recent quantum-resistant KP-ABE schemes [24,41–51] is mainly directed at lattice-based cryptography, which offers security proofs based on NP-hard problems with average-case to worst-case hardness. Moreover, the inherent linear algebra-based matrix or vector operations make lattice-based ABE can be implemented efficiently [15]. Most of these lattice-based ABE schemes [24,41–44,48–51] reduced the hardness of worst-case SVP to average-case LWE problems. However, LWE is known to suffer from inherent quadratic key sizes and computation, thus increasing the challenges of achieving practical quantum-resistant ABE schemes. While R-LWE is proven to be hard with the reduction from worst-case approximate SVP on ideal lattices, several researchers [24,45,47] proposed the R-LWE approach to achieve a more practical ABE scheme by addressing the quadratic overhead of the LWE problem.

In the selective threat model, an attacker declares what the challenge ciphertext will be before he is allowed to view the public parameters [49]. Most quantum-resistant KP-ABE threat models are constructed selectively and secured against the IND-CPA [24,41,42,44–50].

On the other hand, the adaptive model allows an attacker to attack during the course of computation based on the information gathered by the attacker so far. Hence, it is more reasonable to use an adaptive security model to provide better and more realistic security protection [27]. However, recent quantum-resistant ABE schemes that can achieve an adaptive model are limited. Pal and Dutta's [51] scheme enjoys stronger security than the recent quantum-resistant ABE schemes [24,41,42,44–50], in which the adversary is allowed to query the secret keys that can retrieve the challenge ciphertext. Collusion resistance is important in preventing unauthorized users from working together to combine their attributes to access the data illegally. However, recent quantum-resistant [24,41,42,44–51] did not prove their scheme secure under collusion attacks. Collusion resistance is not particularly being implemented into the schemes as well as the revocation mechanism. Future construction of quantum-resistant KP-ABE schemes should include analysis to resist collusion attacks from unauthorized users and revoke any users who no longer own the eligibility to access data. Revocation mechanisms can be generally performed at three levels, including user-level revocation, attribute-level revocation and hybrid-level revocation. Most recent quantum-resistant KP-ABE designs do not take into account the perspective of revocation mechanisms. Whereas, Luo et al. [50] directed to address the user-level renovation. Access structure is another important aspect in designing the KP-ABE scheme to determine its sustainability in supporting user scalability. Monotone access structures are widely used in the construction of quantum-resistant KP-ABE structures to support the scalability issues of a practical ABE scheme, in which if A is a set of attributes satisfying an access structure $\tau$, then any A′ such that A⊂A′ also satisfies $\tau$. The monotone access structure can be formulated in the Boolean circuits such as AND, OR, and NOT gates. Subsequently, the Linear Secret Sharing Scheme (LSSS) was a standard conversion tool to support threshold monotonic access structure. On the other hand, Dai et al.'s KP-ABE scheme [24] focused on the non-monotone access structure with NAND gates to improve backtracking attacks and the size of attribute lists.

### 4.2. CP-ABE Schemes

The ciphertext policy attribute-based encryption (CP-ABE) scheme was designed to address the concerns raised by the KP-ABE technique. The private key in a CP-ABE scheme will be labeled with a list of attributes, and a concrete access policy will be linked directly with each ciphertext [52]. Hence, if the user attributes satisfy the ciphertext's access structure, a user can decrypt and access the data. The implementation of the CP-ABE scheme is also based on four algorithms: Setup, Key Generation, Encryption, and Decryption.

- Setup ($k \rightarrow mk, pp$): Takes the security parameter, $k$ as input and outputs a master key, $mk$ and public keys, $pp$.
- Key Generation ($mk, S \rightarrow sk$): Takes the master key, $mk$ and a set of data user attributes, $S$ to produce a secret key, $sk$.
- Encryption ($pp, M, \tau \rightarrow CT$): Takes as input the public keys, $pp$, the message, $M$ and an access structure, $\tau$, and outputs a ciphertext $CT$.
- Decryption ($CT, sk \rightarrow M/\perp$). The decryption algorithm takes as input the ciphertext, $CT$ and the private key, $sk$ and outputs the decrypted data, $M$. The decryption is successful only if the user attributes satisfy the access structure included in the ciphertext. Otherwise, the output is $\perp$.

The CP-ABE scheme supports user scalability [53]. However, it also faces several drawbacks. The ciphertexts of quantum-resistant CP-ABE grow linearly as the number of attributes increases [10]. Moreover, it suffers from a problem called temporal attributes in dynamic environments. For instance, the attributes in eHealth applications may change over time and are not suitable for dealing with an attribute–user revocation without relying on an authority [40]. Table 4 summarizes the recent works of quantum-resistant CP-ABE.

**Table 4.** Survey of Quantum-Resistant CP-ABE Schemes.

| Year | Ref. | Algorithm Design | Access Structure | Hardness Assumption | Threat Model | Ciphertext Indistin-guishability | Collusion Resistance | Revocable |
|------|------|------------------|------------------|---------------------|--------------|----------------------------------|----------------------|-----------|
| 2012 | Zhang et al. [33] | Lattice-based | Threshold n gate | LWE | Selective | IND-CPA | No | - |
|      | Zhang and Zhang [54] | Lattice-based | AND gates on positive and negative attributes | LWE | Selective | IND-CPA | No | - |
| 2014 | Wang [55] | Lattice-based | AND-gates on multi-valued attributes | LWE | Adaptive | IND-CCA | No | - |
| 2015 | Fun and Samsudin [56] | Lattice-based | LSSS | R-LWE | Selective | IND-CPA | Yes | - |
|      | Zeng and Xu [57] | Lattice-based | AND gate | LWE | Selective | IND-CPA | No | - |
| 2016 | Tan [58] | Lattice-based | LSSS | R-LWE | Selective | IND-CPA | Yes | - |
| 2017 | Fun and Samsudin [59] | Lattice-based | LSSS | R-LWE | Selective | IND-CPA | Yes | - |
|      | Chen et al. [60] | Lattice-based | Threshold n gate | R-LWE | Selective | IND-CPA | No | - |
| 2019 | Yang et al. [61] | Lattice-based | Binary Tree | R-LWE | Selective | IND-CPA | No | Attribute-level |
|      | Tsabary [62] | Lattice-based | Threshold n gate | LWE | Adaptive | IND-CCA2 | No | - |
|      | Liu et al. [63] | Lattice-based | Threshold n gate | R-LWE | Selective | IND-CPA | No | - |
|      | Li et al. [64] | Lattice-based | AND gates on positive and negative attributes | LWE | Selective | IND-CPA | No | - |
| 2020 | Affum et al. [65] | Lattice-based | Boolen Threshold N gates | R-LWE | Selective | IND-CPA | No | - |
|      | Zhao et al. [66] | Lattice-based | Threshold N gates | R-LWE | Selective | IND-CPA | Yes | Attribute level |
| 2021 | Qian and Wu [67] | Lattice-based | Access tree with AND and OR gates | LWE | Selective | IND-CPA | Yes | - |
|      | Varri et al. [68] | Lattice-based | LSSS | LWE | Selective | IND-CKA | No | - |

In 2012, Zhang et al. [33] extended Sahai and Water's [53] identity-based encryption concepts to a lattice-based CP-ABE scheme to support multi-valued attributes. Zhang and Zhang [54] studied the CP-ABE scheme in q-ary lattices to support multi-bit operations, however, their scheme suffered from quadratic overhead issues. Wang [55] improved AND-gates to multi-value attributes. Fun and Samsudin [56] solved the computation

overhead of the CP-ABE scheme by extending to the hardness of R-LWE assumptions. However, their shared master key is vulnerable. Zeng and Xu [57] extended the CP-ABE scheme to support keyword-searchable functions; however, it was designed based on the overhead LWE problem. To support the outsourced cloud data computation, Tan and Samsudin [58] extended the CP-ABE with homomorphic encryption with the hardness of R-LWE assumptions. Subsequently, Fun and Samsudin [59] studied CP-ABE in securing the Internet of Things. Chen et al. [60] improved the performance of the R-LWE-based CP-ABE scheme by proposing a small universe threshold CP-ABE scheme. Yang et al. [61] and Zhao et al. [66] enhanced the CP-ABE scheme with a binary tree structure and threshold gates, respectively. Tsabary [62] designed a CP-ABE scheme from t-CNF based on the LWE problem. Liu et al. [63] solved the user scalability issues by extending the threshold access structure to support the multi-authority level. Li et al. [64] solved the proxy re-encryption issues in CP-ABE with trapdoor sampling and vector decomposition techniques. Affum et al. [65] studied the R-LWE-based CP-ABE scheme to support a 5G content-centric network. Qian and Wu [67] enhanced tree structure by proposing a basic access tree (BAT) to express any disjunctive normal form (DNF). Varri et al. [68] extended CP-ABE to searchability over encrypted data; however, their scheme is constructed from the overhead LWE problem.

Similar to KP-ABE, recent quantum-resistant CP-ABE schemes are mainly constructed based on lattice-based cryptography. The hardness assumption of the quantum-resistant CP-ABE schemes [33,54–68] is from average-case to worst-case problems. In order to solve the inefficiency of LWE due to the inherent quadratic overhead problem, R-LWE is implemented in some of the schemes [58–61,63,65,66] to reduce the computation cost and transaction bandwidth. Like KP-ABE schemes, most CP-ABE schemes [33,54,56–61,63–68] only analyzed their threat model secure against the selective approach and proven IND-CPA secure. Only Wang [55] and Tsabary [62] CP-ABE schemes were tested under the adaptive model to achieve IND-CCA. On the other hand, CP-ABE schemes that are designed to be collusion resistant are very limited. Most recent CP-ABE schemes [33,54,55,57,60–65,68] did not prove the proposed schemes are secure against collusion attacks among multiple authorized users.

The access structure of recent quantum-resistant CP-ABE schemes was mainly constructed based on Boolean tree, AND and OR gates, LSSS and threshold gate. They are known as monotone access structures without negative attributes and are being implemented according to the compatibility of the schemes. Whereas Zhang and Zhang [53] and Li et al. [63] further extend the AND gates to support positive and negative attributes. Subsequently, most recent works did not further design their CP-ABE schemes to support user level or attribute level revocation during real-world implementation. Only Yang et al. [60] and Zhao et al. [65] embedded the attribute level revocation in their CP-ABE schemes.

## 5. Ongoing Challenges and Future Suggestions

While ABE promises a solution to address the scalability issues of recent public key infrastructure (PKI), how the quantum-resistant ABE schemes can be further extended and cope well to support the real-world implementation of big data analytics with its characteristics of high volume, high velocity and high velocity, have attracted numerous researches proposals [32,69–77]. The ABE scheme is a suitable access control method that secures the private and sensitive big data stored in the cloud. Data owners are given the right to control the authorized users who are able to decrypt the data. Chandrasekaran and Balakrishnan [67] suggested an ABE scheme for big data technology in the cloud; however, their scheme is constructed based on attribute union and quadratic residue with a fundamental arithmetic theorem. The scheme is said to be efficient because the probability for a set of user attribute to occur as a squared value is lower. Then, an access control technique based on the KP-ABE scheme and identification system is proposed. Before accessing data from cloud service providers, end-users must verify themselves, and this technique employs a third party to manage key exchange protocol independently from

the cloud service providers. This is because, in the KP-ABE scheme, the ciphertext cannot choose who can decrypt it [32]. Encryption techniques are also used in [65], where the collusion resistance is the main goal and simultaneously allows users to share data among authorized users at the fine-grained level.

However, the enforcement of the privacy protection act and data regulations to protect individuals' privacy could backfire someday. For example, data that are collected and retained for a couple of years can cause privacy violations as long as one keeps the data. Too many data policies and regulations can also hinder innovation if data must be kept as it is without being manipulated. Therefore, one of the challenges in ensuring security and privacy when dealing with big data is to come up with an approach that covers the regulations and analytics. In other words, how the technology can be secured and maintain its high efficiency remain open research questions. Implementing ABE schemes into big data technology also required good management of strategies. Access methods and query processes of big data management have to be secured [73]. The issue becomes more challenging when various policies in ABE schemes need to be applied to heterogeneous data structures to support the high velocity of big data processing. Therefore, the implementation of ABE schemes in the management of big data needs to be examined closely to make sure that the policies can be handled without affecting the performance of the big data technology.

ABE schemes in securing big data do not guarantee that the technology is safe from quantum attacks. Recent quantum-resistant ABE schemes [41–68] mainly constructed based on lattice-based cryptography could be a double sword in big data technology. The quantum-resistant ABE scheme built based on the hardness of LWE assumption with high computation cost due to its inherent quadratic key sizes might have efficiency problems that can affect big data's performance in a real-world implementation. The challenging and open research problems of quantum-resistant ABE schemes in protecting big data are further pointed out as the necessity for further study to develop practical quantum-resistant ABE technologies as below:

- **Efficient quantum-resistant ABE schemes without lattice-based cryptography**. Recent quantum-resistant KP-ABE schemes [41–51] and CP-ABE schemes [33,54–68] are primarily constructed based on lattice-based cryptography. While lattice-based cryptography has only been secured in the inefficiency of large dimensions, it is intriguing to further investigate whether a practical quantum-resistant ABE scheme may be built from other quantum algorithms, such as supersingular elliptic curve isogeny or multivariate approach.

- **Scalability and complexity of access control policy**. Managing high data volume and expandable users is challenging in big data technologies. While quantum-resistant CP-ABE schemes [33,54–68] better support the users' scalability than KP-ABE schemes [41–51], designing attribute directly revocable LSSS threshold gate access policies with backward and forward secrecy is still challenging. The access structure design should consider that users might frequently change in the group, and the policies and keys should be able to be updated timely. Backward secrecy ensures the newly joined users cannot read any previously encrypted data until the data are re-encrypted with the updated attribute key. In contrast, forward secrecy provides that revoked users must not be able to read any future encrypted data till the next expiration. Instead of periodic and scheduled revocation, most recent quantum-resistant ABE schemes [61,66] focused on immediate attribute revocation. Luo et al. [50] employs concepts of proxy re-encryption, which allows semi-trusted proxies to re-encrypt data with the updated access structure. However, proxy re-encryption cannot practically support distributed applications in big data processing, and the risk of collusion attacks in semi-trusted environments needs to be further addressed.

- **Adaptively secured Quantum-resistant ABE Schemes**. As aforementioned, the adaptive threat model is more robust than a selective model since the challenge attribute can be selected whenever even after the attacker has obtained the public param-

eters and decryption keys. Existing adaptively secured quantum-resistance ABE schemes [33,54,55,57,62,64,67,68] are mainly designed based on the hardness of LWE problems that suffer from practical issues, including high dimension cost to embedded expressive access policies. While the R-LWE approach promises a more practical solution to lattice-based ABE schemes, it is necessary to design adaptively secure ABE schemes to assure realistic security in big data technologies.

- **Collusion-resistant ABE schemes**. In collusion-resistant ABE schemes, users cannot combine their attributes to recover the encrypted data. The LSSS widely used in constructing monotone access structures still suffers from collusion attacks and high storage costs. The hardness of LSSS assumptions is based on $(t-1)$ users will not collude; however, it cannot assure if at least $t$ users collude. In distributed big data storage and processing, the possibility of collusion between the semi-trusted service providers that hold the shares secret is very high. While recent collusion-resistant ABE schemes focus on analyzing the collusion resistance among the revoked users, it is necessary to analyze the designed schemes against the collusion among authorized users to recover the secret key.

## 6. Conclusions

This survey discussed the attribute-based encryption schemes from the post-quantum point of view. A gap analysis of the existing reviews and surveys on ABE schemes based on the quantum perspective and applications is presented. Subsequently, a comparison of the conventional pairing-based ABE schemes with the quantum-resistant ABE schemes by analyzing algorithm constructions, computational hardness assumption, storage efficiency, professor efficiency and transmission bandwidth efficiency. Quantum-resistant ABE schemes enjoy robust security to withstand quantum attacks while at the same time preserving the features of ABE schemes in providing fine-grained access control, scalability, flexibility, and data confidentiality. Subsequently, we summarized the criteria of an ideal quantum-resistant ABE scheme that is necessary to be fully adapted to supporting real-world big data applications. Next, a technical review of recent quantum-resistant ABE schemes is divided into KP-ABE and CP-ABE schemes, including their algorithm design, access structure, hardness of security assumptions, threat model, ciphertext indistinguishability, collusion resistance and revocable are presented. While recent quantum-resistant is mainly constructed from the hardness of lattice-based cryptography, we further point out the ongoing challenges and future works of quantum-resistant ABE schemes in securing big data processing.

**Author Contributions:** Conceptualization, S.F.T. and Z.B.J.; methodology, S.F.T.; validation, S.F.T. and Z.B.J.; formal analysis, Z.B.J.; investigation, Z.B.J. and S.F.T.; writing—original draft preparation, S.F.T. and Z.B.J.; writing—review and editing, S.F.T. and G.-C.C.; visualization, S.F.T. and G.-C.C.; supervision, Z.B.J.; project administration S.F.T.; funding acquisition, S.F.T. All authors have read and agreed to the published version of the manuscript.

**Funding:** This research was funded by the Universiti Malaysia Sabah, Research Grant SDK0165-2020.

**Institutional Review Board Statement:** Not applicable.

**Informed Consent Statement:** Not applicable.

**Data Availability Statement:** The data presented in this study are available on request from the corresponding author. The data are not publicly available due to the data involving confidential information of our research group.

**Acknowledgments:** This research was supported by the Universiti Malaysia Sabah Research Grant-SDK0165-2020.

**Conflicts of Interest:** The authors declare no conflict of interest.

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
