# Peer review of "Attribute-Based Encryption in Securing Big Data from Post-Quantum Perspective: A Survey"

_cryptography, doi:10.3390/cryptography6030040_

Round 1

Reviewer 1 Report

In this paper, authors have reviewed some of the existing quantum-resistant attribute-based
encryption. Also current challenges and futures directions are given. Following are my
suggestions.
1. At some places in the text ‘the ABE’is used while at some other place sentence(s) are
started with ABE. If possible, make it uniform.
2. In the introduction, references may be given. For example after first sentence of the
introduction.
3. All the surveys given in Tables 3 and 4 are very recent. What are the advantages of
this review?
4. Page 2 line 52, ‘[10]focused’ put space between [10] and focused
5. Page 4, the following sentence is not very clear ‘In order to 130 create an ideal ABE
scheme, several design requirements a quantum-resistant ABE scheme 131 must fulfil
are summarized as follows.’
6. Page 4, ‘The design of quantum-resistant ABE should’ should be ‘The design of
quantum-resistant ABE schemes should’
7. Page 5, line 180: A ideal should be an ‘ideal’
8. Page 7, line 256: ‘three levels,.including’ I think full stop should be removed after the
,. Also the next sentence should be rephrased.
9. Page 11, line 377: ‘it is necessary to analysis’ may be replaced with ‘it is necessary to
analyse ’

Author Response

Dear Chief of Editor and Reviewer,

We are pleased to resubmit the revision of our manuscript: cryptography-1818447 “Attribute-based Encryption in Securing Big Data from Post-Quantum Perspective: A Survey”.

We deeply appreciated the constructive comments and valuable suggestions from reviewers.

Subsequently, we have addressed each of the reviewer's concerns as outlined in the attached file.

Reviewer 2 Report

This paper is a review of attributed-based encryption, the paper concentrates on this area.

The paper presents an overview of an interesting subject however for clarity and completeness to the reader, I have the following comments in order that the paper would be acceptable:

-The authors should discuss in detail the general subject of cryptography and quantum cryptography and some well-known algorithms could be included for pedagogical purposes.

-The authors should give some examples of the practical algorithms in attributed-Based Encryption.

- Practical examples need to be included in each section.

-The authors wrote “Monotone access 260 structures are widely used in the construction of quantum-resistant KP-ABE structures, i.e. AND, OR, and threshold gates that can be presented in LSSS and Boolean circuits. Cryptography 2022, 6, x FOR PEER REVIEW 8 of 14

Only Dai et al. KP-ABE scheme [22] focused on the non-monotone access structure: AND,  OR and NAND gates. “

This should be clarified in detail.

- Limitations should be discussed in detail with examples.

Author Response

(The authors gave the same response as above.)

Reviewer 3 Report

1.    Relevant research background needs to be supplemented in the abstract.

2.    You should quote all the papers you use correctly. It is recommended to cite references: Yulin Ma, Nachuan Li, Wenbin Zhang, Shumei Wang, and Hongyang Ma, "Image encryption scheme based on alternate quantum walks and discrete cosine transform," Opt. Express 29, 28338-28351 (2021)

3.    In this paper, lack the conclusion analysis of the table, which needs further analysis.

4.    It is recommended to focus on the differences between each scheme and the characteristics of this article

5.    Please add some references.

6.    The statement in Table 4 is not detailed enough.

7.    Inconsistent spacing between title paragraphs.

Author Response

(The authors gave the same response as above.)

Round 2

Reviewer 2 Report

The revised version is ok. I accept it.

Reviewer 3 Report

 Accept in present form